# Uncovering SARS-CoV-2 Molecular Epidemiology Across the Pandemic Transition: Insights into Transmission in Clinical and Environmental Samples

**DOI:** 10.3390/v17050726

**Published:** 2025-05-19

**Authors:** Vrushali D. Patil, Rashmi Chowdhary, Anvita Gupta Malhotra, Jitendra Singh, Debasis Biswas, Rajnish Joshi, Jagat Rakesh Kanwar

**Affiliations:** 1Department of Biochemistry, All India Institute of Medical Sciences, Bhopal 462020, India; vrushali.phd2022@aiimsbhopal.edu.in (V.D.P.); jagat.biochemistry@aiimsbhopal.edu.in (J.R.K.); 2Department of Translation Medicine, All India Institute of Medical Sciences, Bhopal 462020, India; anvitagupta16@gmail.com (A.G.M.); jitendra.tmc@aiimsbhopal.edu.in (J.S.); 3Department of Microbiology, All India Institute of Medical Sciences, Bhopal 462020, India; debasis.microbiology@aiimsbhopal.edu.in; 4Department of General Medicine, All India Institute of Medical Sciences, Bhopal 462020, India; rajnish.genmed@aiimsbhopal.edu.in

**Keywords:** coronavirus, environmental surveillance, next-generation sequencing, outbreak prevention, silent circulation, immunocompromised patients

## Abstract

Background: Respiratory droplets are the main way in which the COVID-19 pandemic’s causal agent, severe acute respiratory syndrome coronavirus-2 (SARS-CoV-2), spreads. Angiotensin-converting enzyme 2 (ACE2) receptors, especially in lung cells, allow the virus to enter host cells. However, ACE2 expression in intestinal cells has sparked worries about possible fecal transfer, particularly in poor-sanitation areas like India. Methods: Between July 2021 and July 2024, clinical (nasopharyngeal, saliva, and stool samples) and sewage samples were collected from outpatient departments and sewage treatment plants (STPs), respectively, from the high-population-density area under study in order to investigate SARS-CoV-2 transmission. Results: This proof-of-concept study analyzed clinical samples from *n* = 60 COVID-19-positive patients at a central Indian tertiary care hospital and *n* = 156 samples from hospital STPs. Variants of SARS-CoV-2 were found using qRT-PCR and Next-Generation Sequencing (NGS). Of the *n* = 37 qRT-PCR-positive patients who gave their assent, 30% had stool samples that tested positive for viral RNA. In 70% of positive NP and 65% of positive saliva samples, along with two stool samples from immunocompromised patients, the live virus was identified using Vero E6 cell lines. Although 18% of the tests reported qRT-PCR-positive results, no live virus was detected in sewage samples despite NGS validation. The detection of SARS-CoV-2 in the absence of confirmed clinical cases may indicate the silent circulation of the virus within the community, suggesting that sewage surveillance can serve as an early warning system before an outbreak occurs. Conclusions: These findings provide critical insights into the importance of continuous environmental surveillance, silent virus circulation, changes in viral epidemiology throughout the years, and strategies to mitigate coronavirus outbreaks.

## 1. Introduction

As a highly contagious viral disease, coronavirus disease 2019 (COVID-19) is spread by the positive-stranded ssRNA virus severe acute respiratory syndrome coronavirus 2 (SARS-CoV-2). Over 6 million people have died as a result of COVID-19, which has had a devastating impact on the world. The virus spread rapidly across the globe after the first case of disease was reported in Wuhan, Hubei Province, China, in late December 2019. On March 11, 2020, the World Health Organization (WHO) declared it a global pandemic [1]. The SARS-CoV-2 virus attaches to host cells through several receptors, including antibody–FcR complexes, neuropilin-1, tyrosine-protein-kinase receptor UFO (AXL), and angiotensin-converting enzyme 2 (ACE2) [2]. ACE2 is expressed throughout the body, with significant concentrations in the lungs and intestinal enterocytes. As a result, patients often experience gastrointestinal symptoms in addition to fever, cough, cold, and a loss of taste and smell [3,4].

The population of asymptomatic individuals is a significant contributor to the propagation of COVID-19. According to studies, pre-symptomatic infections account for approximately 48.9% of cases, and 15.6% of patients are still asymptomatic at the time of initial testing [5]. By promoting the development of herd immunity, COVID-19 immunizations were essential in the battle against the illness [6,7,8]. Despite tremendous progress in clinical research, SARS-CoV-2 variant outbreaks are reported all over the world. In late December 2020, Alpha (B.1.1.7), the first variant of concern, was discovered in the UK [1,9,10]. Omicron and its descendant JN.1, and its subvariants KP.2 and KP.3, have shown substantial prevalence in the United States (JN1 variant) [9,10,11,12].

Effectively monitoring viral circulation is essential due to the frequent occurrence of viral outbreaks. People may be discouraged from getting tested because of the discomfort associated with traditional COVID-19 testing methods, such as nasopharyngeal swabs, particularly for older adults and adolescents, using RT-PCR. SARS-CoV-2 is a positive-sense single-stranded RNA (ssRNA) virus. Understanding this is crucial for effectively isolating and amplifying its genetic material. Since DNA is more stable than ssRNA, it requires special care to minimize RNase activity during RNA extraction. Additionally, reverse transcription is typically required to convert RNA into complementary DNA (cDNA) before performing PCR.

Saliva collection, on the other hand, is a reliable substitute for swab-based methods, is less intrusive, and may be carried out on one’s own without medical help [13,14]. Recent studies [15,16,17] have shown that SARS-CoV-2 can persist in patient stool samples for up to 2–4 weeks. In many cases, COVID-19 patients’ feces and urine are released into sewage systems, which eventually flow into wastewater treatment facilities. This process represents a significant route for SARS-CoV-2 transmission through water and wastewater systems. The surveillance of sewage for SARS-CoV-2 offers several advantages over individual stool sample testing. For one, sewage surveillance provides a more comprehensive overview of infection patterns at the population level, identifying trends in entire communities rather than just in individual patients. In fact, monitoring one liter of wastewater is equivalent to screening a population of two hundred thousand people [18].

Since sewage testing can detect the presence of SARS-CoV-2 before clinical cases manifest, this approach is also beneficial for the early detection of viral epidemics [19]. Sewage surveillance is also an accurate and efficient method of determining virus prevalence because it is non-invasive and does not depend on volunteer involvement. Sewage surveillance offers a more comprehensive picture of the actual illness burden in a community by identifying people who are asymptomatic or have minor symptoms but might not seek medical help [20]. Additionally, up to 14–41 days before clinical cases are reported, sewage surveillance can detect the presence of viruses, serving as an early warning system [21,22,23,24,25,26,27]. This capability allows public health authorities to respond proactively to outbreaks before widespread clinical cases emerge [28]. Considering all of these factors, sewage testing for SARS-CoV-2 is a powerful tool for pandemic surveillance and can complement traditional clinical testing methods.

This study aims to perform the molecular identification and epidemiological analysis of SARS-CoV-2 infection in nasopharyngeal, saliva, stool, and sewage samples, along with establishing the transmission lineage of COVID-19 cases detected in various regions of Bhopal, Madhya Pradesh, India.

## 2. Materials and Methods

### 2.1. Study Design

In this study, COVID-19-positive patients’ nasopharyngeal, neat saliva, and stool samples were collected from the OPD, IPDs, and wards of a tertiary care hospital. Stool samples were collected again on the 7th, 14th, and 21st days. The majority of the patients provided their stool samples only once after they tested positive. Simultaneously, weekly sewage samples were collected from the hospital sewage treatment plant (*n* = 52 samples/year) (Figure 1 and Figure 2).

#### 2.1.1. Clinical Specimen Collection from Patients with COVID-19

Demographic and clinical data of the patients, such as medical history, age, gender, co-morbidities, signs, symptoms, and immunization status, were collected. A list of laboratory investigations of all enrolled patients was also obtained from the record system of the COVID-19 OPD, in the Medicine Department of the institute.

#### 2.1.2. Sewage Sampling Sites

The sewage samples were collected from the sewage treatment plant (STP) of the AIIMS hospital. The STP catchment area includes the AIIMS hospital and campus hostel. The capacity of the STP is 1 MLD (million liter per day). In Figure 2, details of population density and the sample collection site with the highest number of study participants visiting the AIIMS hospital are shown.

### 2.2. Sample Processing

(i)Nasopharyngeal samples and neat saliva: samples were collected from COVID-19 patients with the CDC-recommended procedure [29].(ii)Stool sample processing: Stool samples were processed following a protocol adapted from [30] with some modifications. Briefly, 10 mL of Phosphate-Buffer Saline (PBS), 1 g of glass beads, and 1 mL chloroform (CHCl3) were added to each tube. Approximately 2 g (peanut size) of stool sample was transferred to the labeled centrifuge tube. This was carried out inside a biosafety cabinet (BSC) in a biosafety level 2+ (BSL2+) laboratory. The centrifuge tubes were tightly secured and shaken (mixed) vigorously on the mechanical shaker for 20 min (min) for the proper mixing of stool content and reagents. After shaking, the tubes were centrifuged (refrigerated centrifuge) at 3000 rpm (1500× *g*) for 20 min. The supernatant was further filtered by using a 0.45 µm syringe filter. The filtered stool supernatant produced was stored at −20 degrees Celsius until testing.(iii)Sewage sample processing: Every week after 10:00 a.m., 200 mL of sewage samples (both influent and effluent) was taken from the AIIMS STP using the grab sampling technique and placed in sterile glass bottles. The bottles were carried on a cooled chain after being cleaned with 2% sodium hypochlorite. Samples were processed at a BSL2+ facility [31]. Two 40 mL replicas were created, one of which was heat-inactivated for 90 min at 60 °C, and the other was kept at −80 °C. After PEG (8000 MW) and NaCl were added, the mixture was vortexed and centrifuged for 40 min at 4 °C at 10,000 rpm. The pellet of the heat-inactivated aliquot was resuspended in lysis buffer, and since SARS-CoV-2 is an RNA virus, precautions were taken to ensure RNA stability and reduce RNAse activity. In view of this, viral RNA was extracted using the Qiagen Viral RNA Mini Kit, with 50 μL eluate stored at −80 °C for reverse transcription followed by QRT-PCR. Positive samples were further processed, and pellets of the non-heat-inactivated aliquot were resuspended in 1× PBS for live virus screening on a Vero E6 cell line.

### 2.3. Virus Isolation

The SARS-CoV-2 qRT-PCR-positive clinical and environmental samples were used for virus isolation, for its growth over Vero E6 cell lines maintained in minimum essential media (MEM) containing 10% Fetal Bovine serum (FBS) along with 1% of Penicillin–Strep9tomycin (10,000 U/mL) solution and 0.25 µg/mL Amphotericin B, and viral infection was achieved using a method mentioned in [16,32]. After the inoculation of nasopharyngeal samples, the culture was observed for a cytopathic effect (CPE) for 5 days and subcultured after 5 days, and in the case of stool samples, the cultures were observed daily for a CPE for 7 days. After this incubation, a blind passage was performed in all samples, and the presence of SARS-CoV-2 was confirmed by qRT-PCR in the supernatant of the cell cultures followed by genomic sequencing. A similar protocol was performed for the sewage samples as well (as shown in Figure 3).

### 2.4. Whole-Genome Sequencing and Mutational Analysis

Total RNA was extracted from nasopharyngeal and oropharyngeal swab samples using the QIAamp^TM^ Viral RNA Mini Kit, following the manufacturer’s protocol. The purity and concentration of RNA were assessed using a Qubit 4 Fluorometer. cDNA synthesis was performed using the SuperScript™ IV First-Strand Synthesis System (Thermo Fisher, Waltham, MA, USA) with random hexamers. Whole-genome amplification was carried out using the Ion AmpliSeq™ SARS-CoV-2 Research Panel, which consists of two primer pools targeting the complete viral genome. The amplicons were then purified and subjected to library preparation using the Ion AmpliSeq Library Kit 2.0, followed by quantification with the Qubit dsDNA HS Assay Kit. Template preparation, enrichment, and chip loading were automated using the Ion Chef System, and sequencing was performed on the Ion Torrent S5 Plus System using a 540 chip with a run time of approximately 2.5 h. Raw sequencing reads were processed using Torrent Suite™ software version 5.18 for quality filtering and alignment to the SARS-CoV-2 reference genome (NC_045512.2), and variant calling was performed via Ion Reporter™ software. Lineage classification and phylogenetic analysis were conducted using Nextclade and Pangolin. Quality control metrics, including total reads, read depth, and genome coverage (≥90% at 30× depth), were assessed to ensure high-quality sequencing data suitable for downstream analysis.

## 3. Results

### 3.1. Clinical Sample Results

During the period of July 2021–July 2024, *n* = 60 COVID-19-positive patients were enrolled in the study, comprising 33 (55%) males and 27 (45%) females (as shown in Figure 1 and Table 1) with a median age of 45 years, and after their consent was obtained, their nasopharyngeal, saliva, and stool samples were collected. Comparable qRT-PCR results (as shown in Table 2) were observed in the nasopharyngeal and neat saliva samples.

### 3.2. Environmental (Sewage) Sample Results

#### Total Number of Sewage Samples Collected

During the course of the study, n = 156 sewage samples (Inlet and Outlet) were collected, followed by processing, qRT-PCR, and genomic sequencing analysis. Out of these, *n* = 28 sewage samples (Inlet and Outlet) tested positive with a higher Ct range between 29 and 36, also confirmed through genomic sequencing.

### 3.3. Epidemiological Analysis of Variants Found During Study (2021–2024)

Figure 4 shows significantly important information about the temporal and geographical spread of the SARS-CoV-2 virus. During this study, virus monitoring was successfully conducted using clinical and environmental samples collected from the hospital and the sewage treatment plants (STPs) located at AIIMS Bhopal, respectively. The SARS-CoV-2 Delta variant, which was first reported in India in October 2020, took around 4 months to spread to other countries like South Africa, Bangladesh, Indonesia, USA, Russia, South Africa, etc., from India. In this study, we observed that the Delta variant was detected in sewage surveillance samples on 4 January 2021, and after a month, in the period of 26 February–31 May 2021, clinical cases started to be reported in the study area. Similarly, in the sewage samples collected on 7 June 2022, SARS-CoV-2 was detected, and COVID-19 cases were reported in the study area approximately a month later on 10 July 2022. The sequencing of environmental and clinical samples revealed that the virus belonged to the BA.75 variant (Figure 4).

A similar pattern emerged in 2023 and 2024, where SARS-CoV-2 RNA was detected in sewage samples 30 to 60 days prior to the appearance of clinical cases in the study area. During this period, XBB was identified in sewage and JN.1 variants were identified in both clinical and sewage samples. Notably, viral RNA was detected in sewage even when no cases were reported in hospitals, indicating the silent circulation of the virus. This underscores the value of wastewater surveillance in monitoring viral spread. The presence of viral RNA in sewage highlights not only the capability of surveillance systems to track viral mutations but also their role in detecting silent viral circulation.

This study reinforces the consistent detection of SARS-CoV-2 RNA in both clinical and environmental samples. Notably, nasopharyngeal and neat saliva samples showed comparable results, while viral RNA in stool samples suggested prolonged viral shedding, pointing to a potential fecal–oral transmission route. The early detection of viral RNA in sewage serves as a powerful tool for establishing an early warning system, enabling timely responses to potential outbreaks.

### 3.4. Detection of Live Virus in Clinical Samples

Sixty qRT-PCR-positive nasopharyngeal samples and neat saliva samples were inoculated onto Vero E6 cell lines grown in T-25 flasks, and the cytopathic effect (CPE) was observed. Similarly, 60 stool samples and 74 sewage samples were inoculated onto the same cell lines. Within 4–5 days of inoculation, a CPE was detected in nasopharyngeal and neat saliva samples, followed by subculture, qRT-PCR, and genomic sequencing. Fifteen days post-inoculation, during the second subculture, two stool samples showed a CPE, characterized by clusters of ruptured cells forming aggregates, in contrast to the negative control (Figure 5). All other inoculations remained negative through the fifth subculture. A qRT-PCR analysis (as shown in Table 3) of the supernatants from these two stool samples confirmed active SARS-CoV-2 proliferation, with cycle threshold (Ct) values of 23.2 and 25.6. The corresponding stool samples had previously shown PCR Ct values of 33.2 and 35.4, respectively. Additionally, the 60 stool samples from COVID-19-positive patients were monitored for a CPE.

We could not observe any CPE in sewage samples, but the cell culture supernatant tested qRT-PCR-positive, indicating the virus in sewage samples, with its presence confirmed through genomic sequencing. Whole-genome sequencing of the cell supernatants revealed that they belonged to the BA.2.75 and JN. 1 variants, respectively.

Out of the *n* = 60 collected nasopharyngeal and *n* = 37 stool samples, it was observed in the in vitro study that all nasopharyngeal samples and 2 stool samples showed a CPE (as shown in Figure 5); however, none of the sewage samples showed a CPE. Detailed study of the samples revealed that the samples belong to immunocompromised patients (as shown in Table 4). This crucial observation suggests that the SARS-CoV-2 virus can be circulated in immunocompromised patients and that they can be potential carriers of the live virus after COVID-19 infection. However, the sewage sample study could not detect any live viruses.

### 3.5. Phylogenetic and Mutational Analysis

In total, *n* = 18 samples were sent for sequencing. Detailed mutational analysis was conducted using *Next Clade software* version 3.13.1 [35]. On comparing the mutations that occurred in various genes of the viral genome, it was found that the S-gene had the most prominent mutations, followed by ORF1b (as shown in Figure 6). When these sequences were analyzed using phylogeny, a detailed evolutionary relationship between the various variants was observed. Also, the sequences isolated from clinical and environmental samples were perfectly clustered with the other sequences of SARS-CoV-2 submitted to NCBI, showing the similarity of this study’s sequences to other sequences; additionally, these sequences were clustered within the known variants of SARS-CoV-2, showing the divergence of the virus reported each year (as shown in Figure 7).

## 4. Discussion

Environmental surveillance is an information-based activity associated with large volumes of data, involving their collection, processing, analysis, and interpretation. This is mainly carried out to monitor trends in infectious agents and the effectiveness of control measures, and identify high-risk-population areas to target through interventions [3,18]. COVID-19 disease-causing SARS-CoV 2 is conventionally screened using minimally invasive nasopharyngeal sampling as a part of its diagnosis [29]. Apart from nasopharyngeal samples, the virus has been detected in saliva, stool, and sewage samples [13,17,19,24,26]. This study evaluated several approaches for SARS-CoV-2 detection and surveillance from 2021 to 2024 to identify its epidemiological trends. qRT-PCR and sequencing analysis of COVID-19-positive patients’ nasopharyngeal samples along with their respective neat saliva samples revealed comparable results, indicating that neat saliva can be a good, analogous substitute to nasopharyngeal samples for COVID-19, especially for mass screening [24,25]. Previously conducted studies suggested that the presence of SARS-CoV-2 in a patient’s saliva may be due to salivary gland infection. It should be mentioned, though, that saliva specimens include secretions that travel up from the lung by the action of cilia lining the airway or that travel down from the nasopharynx in addition to saliva released by the major or minor salivary glands. Since nasopharyngeal swabs can be uncomfortable for children and elderly populations, this finding may be pertinent to these populations and enhance early detection and lessen transmission during viral outbreaks [24]. During this study, *n* = 60 COVID-19-positive patients consented to providing NP, saliva, and stool samples, which were included in the study with environmental samples from the selected study region. There is limited information in the scientific data on analyzing more than three types of specimens in a single study [27,39]. The status of all nasopharyngeal and saliva samples was confirmed by molecular characterization and in vitro analysis (Figure 5 and Figure 7). However, we obtained maximum positivity through qRT-PCR in stool samples collected after the 7th or 14th day from the onset of symptoms, and no virus was detected in samples collected prior to the 7th day of onset. Viral shedding has been reported from the stool samples of COVID-19 patients in various studies [17,40]. Furthermore, in the in vitro analysis, the live virus could not be detected, but the live virus was detected in the stool samples collected from immunosuppressed COVID-19 patients. Similar observations were reported in one study, where immunosuppression was linked to the persistence of SARS-CoV-2 that could replicate [25]. Both innate immunity and the T-cell-mediated adaptive immune response are critical for the removal and long-term suppression of viral infections, as has been well documented for many of them [23,24]. It is also critical to recognize the link between the severity of illness and existence of co-morbidities and the persistence of SARS-CoV-2 [39]. This may indicate that the intestinal epithelial cell ACE-2 receptor may play a role in the pathophysiology and mechanism of SARS-CoV-2 gastrointestinal infections. SARS-CoV-2 S protein combines with ACE2 to infect the host. The ACE2 protein’s expression is downregulated following viral infection. The primary biologically active component of RAS, AngII, is changed into Ang1-7 by ACE2; once Ang (1Mel7) binds to the Mas receptor and couples with the Gq protein, it possesses anti-inflammatory and anti-remodeling properties. Consequently, the intestinal anti-inflammatory capacity is expected to decline once the virus has infected the host, and immunocompromised patients may act as carriers in the high-risk population [41,42]. A genomic sequencing analysis of nasopharyngeal, saliva, and stool samples identified various circulating SARS-CoV-2 variants [43]. Initially, the Delta variant was reported in clinical samples collected in the year 2021 followed by BA.2 variants like BA.2.75 in the year 2022, and XBB and JN.1 from samples collected in the years 2023 and 2024, respectively. Our study also identified two different SARS-CoV-2 Pangolin lineages, JN.1.1 in nasopharyngeal and saliva samples and JN.1.13 in a stool sample isolated on the 7th day from the onset of symptoms from the same patient. No significant difference was observed in both the lineages as per the sequencing data. One study reported inter-host and intra-host genome variants in fecal samples, as well as in throat swabs and bronchoalveolar lavage fluid, related to the ongoing evolution of SARS-CoV-2 in the human body, based on sequencing [44]. SARS-CoV-2 has been shown to display location-dependent genetic variations across different anatomical regions and also genetic divergence between viral populations in the respiratory and gastrointestinal tracts. Studies attributed this to selective transmission events that occurred during intra-host movement, demonstrating that the genetic variability in gastrointestinal populations was higher. Similarly, significant intra-host genetic variation across a variety of samples and sequencing platforms was reported, suggesting that these variations were not technical inconsistencies but rather biological differences. Furthermore, even in patients with mild symptoms, sustained viral shedding and intra-host evolution have been reported, indicating that the virus can adapt inside a single host. Together, these results provide evidence for our findings and highlight how crucial it is to take intra-host variation into account in understanding SARS-CoV-2′s evolution [45,46,47,48,49].

This study focuses on easy and scalable virus detection and continued environmental surveillance to establish the SARS-CoV-2 virus’s epidemiological link. In view of this, 156 sewage samples were screened, and out of them, *n* = 28 tested positive for SARS-CoV-2 using qRT-PCR, with Ct values ranging from 29 to 36. This supports the efficiency of wastewater surveillance as an early warning system. The detection of viral RNA in sewage confirms silent circulation before the emergence of clinical cases. For instance, according to a research study, the Delta variant was first reported in India in October 2020 [50]. In our study, the Delta variant was identified in sewage samples in January 2021, nearly a month before a significant spike in clinical cases occurred between February and May 2021. Similarly, SARS-CoV-2 RNA was detected in sewage in June 2022 in our study area, with clinical cases appearing a month later. This trend continued with the emergence of new variants such as XBB and JN.1 in 2023 and 2024, respectively. Notably, viral RNA was found in sewage even when no active cases were reported in local hospitals, underscoring the potential of wastewater monitoring to detect silent community transmission and track emerging variants. Research has verified the correlation between reported COVID-19 cases and the concentration of SARS-CoV-2 in wastewater, providing support for the notion that viral levels in wastewater reflect community infection rates [31].

We demonstrate in this pilot study the viability and effectiveness of wastewater-based epidemiology for early virus detection. Apart from symptomatic patients, 35% of COVID-19-positive cases were asymptomatic, indicating silent virus circulation [7]. Wastewater analysis captures data from both symptomatic and asymptomatic individuals, offering a comprehensive view of the virus’s prevalence. This is particularly valuable since asymptomatic carriers can unknowingly spread the virus. Studies conducted previously by our group have carried out the implementation and analysis of new strategies that would help in viral disease identification and management [18,51,52,53,54,55]. In response, the Centers for Disease Control and Prevention (CDC) in the United States have established the National Wastewater Surveillance System (NWSS) to monitor SARS-CoV-2 levels, strengthening the ability to respond quickly to emerging hotspots and virus variants. Previous studies [23,34,51] have observed a decline in the severity and mortality of SARS-CoV-2, despite the continuous emergence of new variants. This trend is likely attributed to increased vaccination rates, innate protection from prior infections, and the virus’s evolution toward lower pathogenicity. Whole-genome sequencing revealed that the XBB and JN.1 variants exhibited mutations in key viral genes, including the spike (S), ORF1b, nucleocapsid (N), and membrane (M) genes. Specific mutations, such as F486S in XBB and L452R, P681R, and D614G in the Delta variant, were found to enhance receptor binding, potentially affecting transmission dynamics. However, these mutations did not result in more severe illness, suggesting that host immunity plays a crucial role in reducing clinical severity [34]. Phylogenetic analysis showed that SARS-CoV-2 sequences from both clinical and environmental samples closely clustered with globally reported strains, reinforcing the virus’s evolutionary trajectory over the past four years (Figure 4), potentially influenced by infections in immunocompromised patients. These findings provide valuable insights into viral persistence, transmission dynamics, and the evolution of SARS-CoV-2 variants. Additional factors such as vaccination and seasonal changes may also contribute. Ongoing genomic surveillance remains critical for tracking emerging variants and assessing their potential impact on public health. This study serves as proof of concept (POC) and was conducted in a tertiary care hospital and its surrounding area, which constrains surveillance to regions with varying population densities, sanitation infrastructures, and access to healthcare. Despite these limitations, it provides crucial insights into the epidemiology of SARS-CoV-2. The large number of samples collected from households represents a broad spectrum of demographic and clinical profiles, capturing variability in viral shedding. Additionally, genomic sequencing capabilities allow for the identification of dominant circulating variants, as well as low-prevalence mutations or recombinant lineages. Geographical variations play a critical role in viral diversity studies, highlighting how different regions contribute to the emergence of new variants [54,55,56,57,58]. This study, one of the first conducted in central India, underscores the silent circulation of SARS-CoV-2—a characteristic feature of enteroviruses [18,51,52], Dengue, and Orthomyxoviruses [56,59]. These findings position the study within the broader environmental surveillance framework, which is essential for tracking the emergence of new coronaviruses.

## Figures and Tables

**Figure 1 viruses-17-00726-f001:**
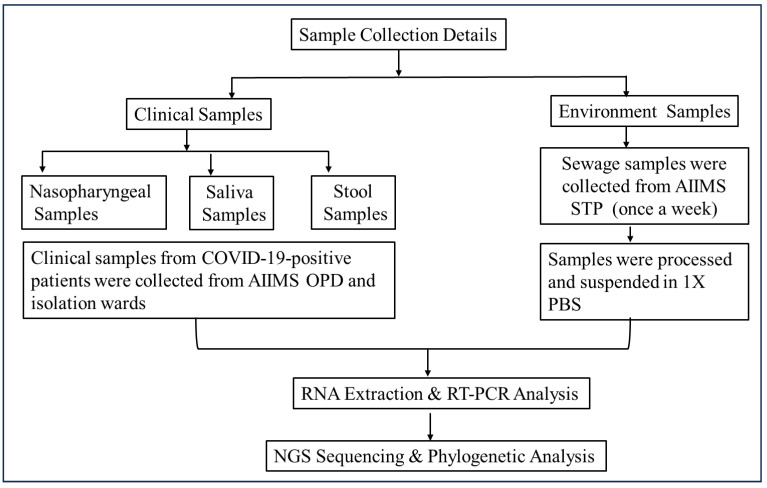
**Flow diagram of sample collection and processing methodology.** Two types of samples (clinical and environmental) were collected to establish the epidemiological link of the virus. Samples were processed and subjected to molecular and genomic analysis using qRT-PCR and Next-Generation Sequencing (NGS), respectively.

**Figure 2 viruses-17-00726-f002:**
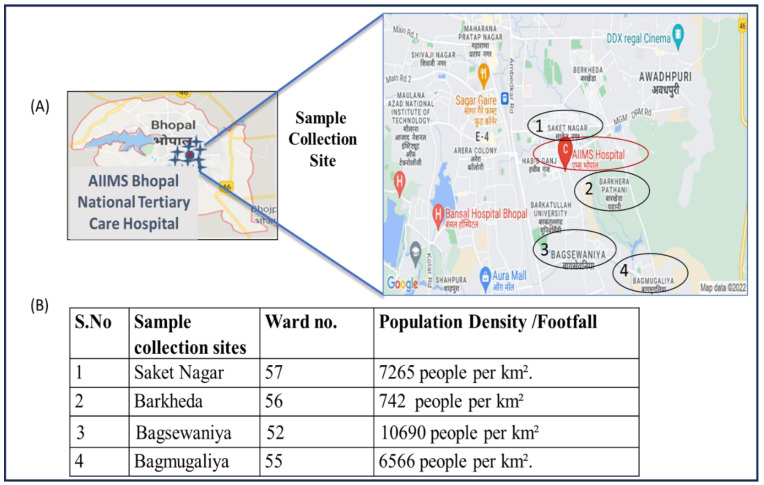
(**A**) Map showing the sites near AIIMS Bhopal with the highest number of study participants visiting the hospital. (**B**) Population density of each ward. During the course of the study, the maximum number of patients was reported from a nearby area of the hospital situated in an approximately 5 km diameter.

**Figure 3 viruses-17-00726-f003:**
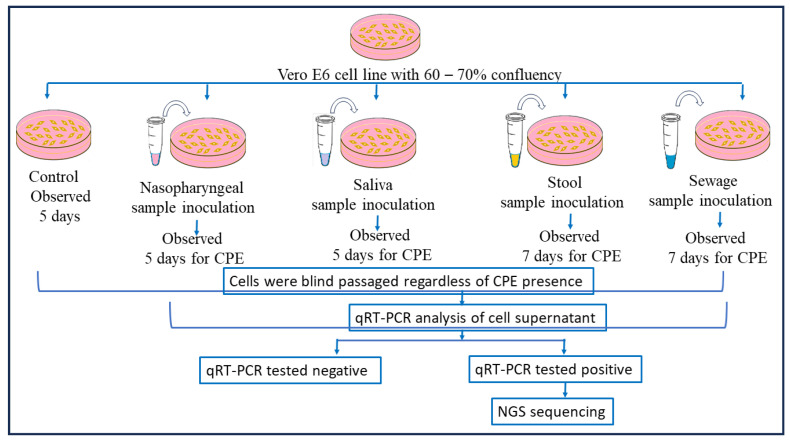
**Detailed virus isolation procedure using nasopharyngeal, saliva, stool, and sewage samples.** Following nasopharyngeal sample inoculation, the culture was monitored for a cytopathic effect (CPE) for five days and then subcultured; for stool samples, the cultures were monitored daily for seven days. Following this incubation, all samples underwent a blind passage, and qRT-PCR was used to validate the presence of SARS-CoV-2 in the cell culture supernatant before genomic sequencing was conducted. The same procedure was followed for the sewage samples.

**Figure 4 viruses-17-00726-f004:**
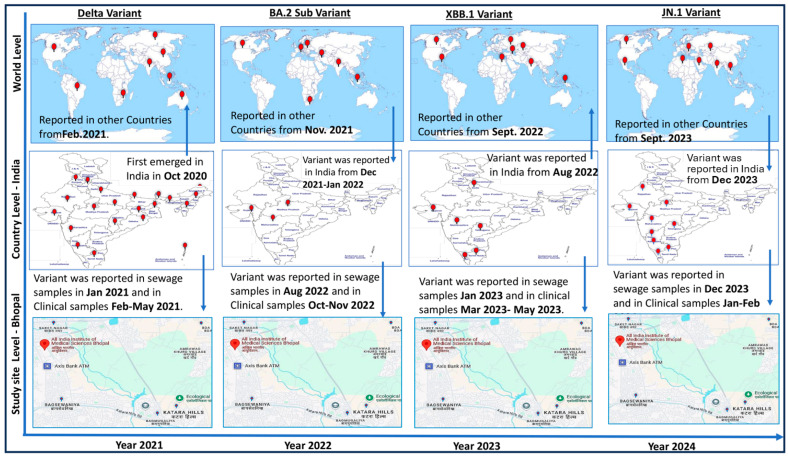
**Year-wise epidemiology of SARS-CoV 2 variants (diagram prepared by author).** The arrows are showing the movement of virus variant from different countries to India and study site and vice-versa at different point of time. Viral monitoring was effectively conducted during the investigation using clinical and environmental samples collected from the hospital and the sewage treatment facilities (STPs) at AIIMS Bhopal, respectively. It took roughly four months for the SARS-CoV-2 Delta strain to move outside of India, where it was first identified in October 2020. On 4 January 2021, this study discovered the Delta variant in sewage surveillance samples. Clinical cases from the study region started to be reported a month later, from 26 February to 31 May 2021. Similar trends were observed in 2023 and 2024, when sewage containing SARS-CoV-2 RNA was discovered 30–60 days before clinical cases [10,31,33,34].

**Figure 5 viruses-17-00726-f005:**
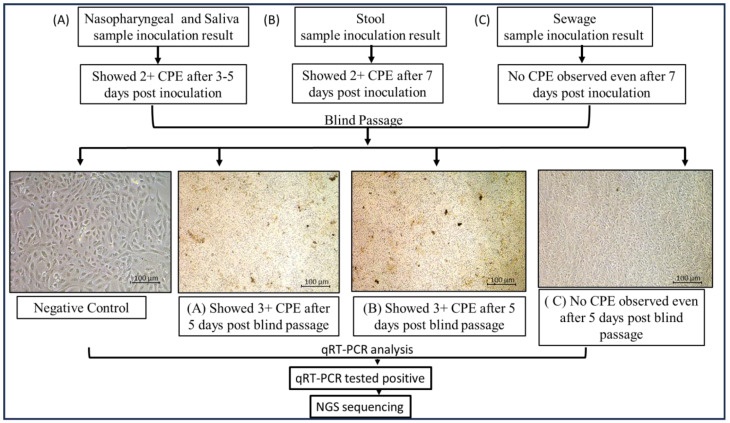
**Cytopathic effect of severe acute respiratory syndrome coronavirus 2 (SARS-CoV-2) on Vero E6 cells inoculated with** (**A**) nasopharyngeal and saliva samples; (**B**) stool samples; (**C**) sewage samples. Cytopathic effect (CPE) was detected after inoculating Vero E6 cell lines cultured in T-25 flasks with 60 qRT-PCR-positive nasopharyngeal samples and neat saliva samples. Likewise, the same cell lines were inoculated with 60 stool samples and 74 sewage samples. CPE was found in nasopharyngeal and clean saliva samples 4–5 days after inoculation. Subculture, qRT-PCR, and genomic sequencing were then performed. In contrast to the negative control, two stool samples from the second subculture, which took place 15 days after inoculation, displayed a CPE, which is defined by clusters of rounded cells forming aggregates (images were taken at 20× using Nikon Eclipse Ti microscope, Nikon Corporation, Tokyo, Japan).

**Figure 6 viruses-17-00726-f006:**
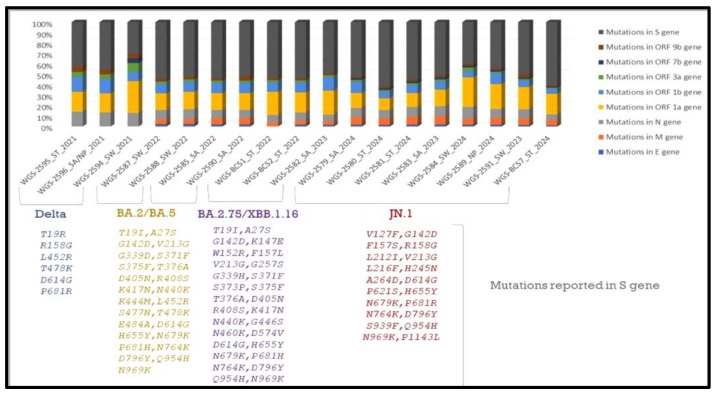
**Detailed mutational analysis of study sequences isolated from clinical and environmental samples.** Samples of the XBB and JN.1 variants were found to have particular mutations in the ORF1b, N, and M genes. Key alterations linked to increased receptor binding affinity were also found in our research, including F486S in the XBB variant and L452R, P681R, and D614G in the Delta variant, in accordance with the literature [35,36,37,38].

**Figure 7 viruses-17-00726-f007:**
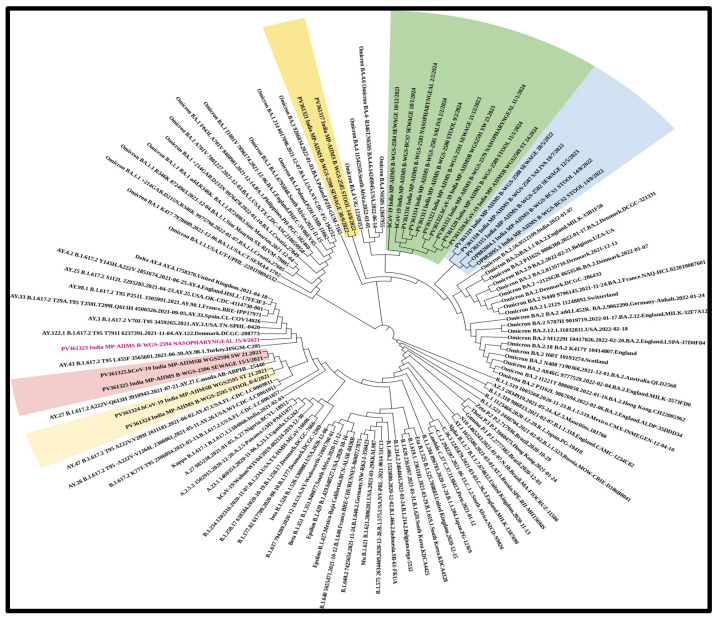
**Phylogenetic analysis of the sequences revealing an elaborate evolutionary link between the different variants.** The sequences extracted from clinical and environmental samples perfectly cluster with the other SARS-CoV-2 sequences submitted to NCBI, demonstrating the study sequences’ similarity to other sequences. These sequences also cluster within the known SARS-CoV-2 variants, indicating the virus’s divergence. The virus was detected in sewage samples prior to in clinical samples. (Samples sequenced as part of the study are highlighted along with the NCBI accession number and collection date: NP—nasopharyngeal; SA—saliva; ST—stool; and SW—sewage sample).

**Table 1 viruses-17-00726-t001:** Baseline characteristics of the COVID-19 patients categorized by sample positivity (*n* = 60).

Patient Details	Values in Percentage
Gender	
Male	33 (55%)
Female	27 (45%)
Symptoms	
Diarrhea	15 (25%)
Fever	42 (70%)
Cough	25 (42%)
Fatigue	20 (33%)
Sore throat	36 (60%)
Headache	11 (18%)
Vaccination status	
Covishield	47 (78.3%)
Covaxin	13 (21.6%)
Co-morbidities/autoimmune disease	
Hypertension or hypotension	12 (20%)
Diabetes (type 1 or type 2)	18 (30%)
Asthma	1 (1.6%)
Rheumatoid arthritis	1 (1.6%)
Cardiovascular disease (CVD)	2 (3.3%)
No co-morbidities	26
Isolation status	
Hospitalized	16 (26.6%)
Home quarantine	44 (73%)

**Table 2 viruses-17-00726-t002:** qRT-PCR results for different samples collected from COVID-19 patients (*n* = 60).

S.No	No. of Specimens Collected	Positive (%)
1	Nasal (*n* = 60)	60 (100%)
2	Neat saliva (*n* = 52)	47 (90%)
3.	Stool (*n* = 37)	11 (30%)

**Table 3 viruses-17-00726-t003:** Details of qRT-PCR-positive samples tested for live virus.

qRT-PCR Positive Sample Type	No. of Positive Samples Inoculated on Cell	No. of Samples Showing Cytopathic Effect	No. of Cell Supernatants Testing qRT-PCR-Positive	No. of Samples Sequenced
Nasal	60	42 (70%)	42	3
Saliva	52	34 (65%)	32	2
Stool	11	2 (18%)	10	6
Sewage	28	0	16	7

**Table 4 viruses-17-00726-t004:** Clinical profiles of patients with stool samples harboring live SARS-CoV-2.

Patient Details	AIIMS/ST/01	AIIMS/ST/02
Age (years)	71 years	61 years
Co-morbidities/autoimmune disease	Rheumatoid arthritis	Asthma
Symptoms with duration	Fever, sore throat, cough, and diarrhea for 15 days	Mild symptoms, including fever, cough, and diarrhea, which lasted for 5 days
Date of stool sample collection	14 November 2022	5 February 2024
Vaccination status	Vaccinated (two doses)	Fully vaccinated (including booster dose)
Isolation status	Hospitalized	Home quarantine

## Data Availability

We have provided the NCBI GeneBank accession numbers of the study sequences in phylogeny (Figure 7).

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
