# Peer review of "Uncovering SARS-CoV-2 Molecular Epidemiology Across the Pandemic Transition: Insights into Transmission in Clinical and Environmental Samples"

_viruses, 2025, doi:10.3390/v17050726_

Round 1

Reviewer 1 Report

Comments and Suggestions for Authors

There were several factors that facilitated the rapid spread of the SARS-CoV-2 virus worldwide, not the least of which was the ability of the virus to be transmitted before symptoms present in infected individuals.  This ability of uninfected individuals to spread the virus, compounded by the fact that the most common method of detection of the virus was nasopharyngeal sampling, a relatively unpleasant technique, further delayed our ability to detect the virus, even after effective testing strategies, i.e. PCR and antibody-based assays, were rapidly developed.  Later in the pandemic cycle, the potential of screening for the virus in wastewater became apparent, as it provides significant advantages in many respects.  These include: 1) detection of the virus prior to the emergence of clinical cases; 2) the ability to detect virus in symptomatic and asymptomatic individuals, resulting in a fuller picture of viral levels; 3) elimination of the need for patient sampling; and, 4) the ability to simultaneously screen a large segment of the population in one sample.

This study provides a persuasive confirmation of the power of wastewater testing in monitoring the levels of SARS-CoV-2 in a population.  The authors present data demonstrating their ability to determine the timing of new variant emergence in the tested population.  Overall, the manuscript is considered exceptionally strong, providing a very concise and clear picture of the transitioning of the virus during the pandemic in this part of the world.  Indeed, it should be stressed that Fig. 4 is considered a major strength, as it deftly presents a quite extensive set of data in an informative, highly illuminating manner that establishes for the reader the epidemiology of the various SARS-CoV-2 variants. 

Despite the multiple strengths identified in the study, there are some weaknesses that diminish the impact of the manuscript.  The following issues must be addressed in a revised version:

  • In line 406, the authors refer to Figure 8 as showing the evolutionary trajectory of the virus. However, I could not locate any such figure in the manuscript.
  • The micrographs showing the cytopathic effects in Vero E6 monolayers inoculated with virus-containing nasopharyngeal, saliva, stool and sewage samples are of quite poor quality and must be improved. Also, in the legend to Fig. 5 (line 270), the CPE in the infected samples are contrasted with the lack of same in the negative control.  But, there is no negative control shown.  This must be included.
  • It is difficult to make a connection between the various panels in Fig. 7 and the conclusions as summarized in the legend to the figure. Please specify that the terminology used in the figure itself applies to each of the B.1.617.2, BA.2, XBB and JN.1 variants.

Author Response

Query 1: In line 406, the authors refer to Figure 8 as showing the evolutionary trajectory of the virus. However, I could not locate any such figure in the manuscript.

Answer 1: We apologize for the typing error. It should be Figure 4. The changes have been made in line 426.

Query 2 : The micrographs showing the cytopathic effects in Vero E6 monolayers inoculated with virus-containing nasopharyngeal, saliva, stool and sewage samples are of quite poor quality and must be improved. Also, in the legend to Fig. 5 (line 270), the CPE in the infected samples are contrasted with the lack of same in the negative control.  But, there is no negative control shown.  This must be included.

Answer 2: Thank You for the valuable suggestion. We have updated the figure 5 (line 274)

Query 3: It is difficult to make a connection between the various panels in Fig. 7 and the conclusions as summarized in the legend to the figure. Please specify that the terminology used in the figure itself applies to each of the B.1.617.2, BA.2, XBB and JN.1 variants.

Answer 3: Thank you for the kind suggestions we have update the phylogenetic tree and figure legends (Page no.7 line 312)

Reviewer 2 Report

Comments and Suggestions for Authors

1) The phylogentic figures in the paper are cladograms with meaningless branch lengths and would be greatly improved by trees with branch lengths proportional to genetic distances.  It would also be nice to add some Alpha, Beta, Delta, Omicron reference sequences (or labeled as B.1.1.7, BA.2.75, JN.1, JN.1.13 etc) so we can see how the Indian sequences fit into the evolution of the virus overall.  One figure, showing the sequences from this study over the years might be more informative than the 4 figures (A - D) taking up so much space in the current views. I am attaching an alignment of "reference sequences" from GISAID plus a few of the new India sequences.  It is easy to delete all but a few of the most relevant reference sequences to build a tree with just the ones closely related to the Indian sequences, and perhaps a few others for contrast. 

2) The study states "Our study also identified two differ-360 ent SARS-CoV-2 Pangolin lineages, JN.1.1 in nasopharyngeal & saliva samples and 361 JN.1.13 in stool sample isolated on the 7th day from the onset of symptoms from the same 362 patient." and seems to imply the virus may have evolved from JN.1.1 to JN.1.13 within this patient, when it seems more likely that this was two different infections of the same patient. 

3) From the methods, it is not clear to me, how stool and sewage samples are treated to potentially allow SARS-CoV-2 virus to survive to infect Vero cells while eliminating bacteria and other viruses that would contaminate the cell cultures.  The paper seems to be implying that wastewater is not a significant risk for spread of the virus, and I agree with that.  But there is possible risk of virus spread via "the fecal - oral route" not from wastewater per se but via lack of handwashing in food handlers, or similar issues.

4) The paper discusses collecting liquid waste water, and does not mention that viral nucleic acids or perhaps whole virions can be captured from waste water on various filter material, effectively concentrating small levels of virus from much greater volumes of water. 

5) The introduction might be improved by noting that this is a positive strand ssRNA virus.  This is not very relevant to the pathology of the virus, but is important for understanding how to isolate and amplify the viral nucleic acid.  DNA is more durable than ssRNA, so precautions need to be used to reduce RNAse activity when isolating RNA from samples, and reverse transcription prior to PCR may be needed. 

Author Response

Query 1: The phylogenetic figures in the paper are cladograms with meaningless branch lengths and would be greatly improved by trees with branch lengths proportional to genetic distances.  It would also be nice to add some Alpha, Beta, Delta, Omicron reference sequences (or labeled as B.1.1.7, BA.2.75, JN.1, JN.1.13 etc) so we can see how the Indian sequences fit into the evolution of the virus overall.  One figure, showing the sequences from this study over the years might be more informative than the 4 figures (A - D) taking up so much space in the current views. I am attaching an alignment of "reference sequences" from GISAID plus a few of the new India sequences.  It is easy to delete all but a few of the most relevant reference sequences to build a tree with just the ones closely related to the Indian sequences, and perhaps a few others for contrast. 

Answer 1: Thank You for your suggestions and providing the alignment of "reference sequences" from GISAID. We have prepared a new phylogenetic tree using the sequences provided by you. Attached on page number 7, line 312. The figure 4 is the crucial part of our study as it provides the information of the global spread of virus and its evolution over time at a glance. As per one of our reviewers it is the major strength of our study. Hence, we would like to keep the image in the manuscript. But as per your valuable suggestion regarding the space of diagram if all the reviewers find it suitable we can create a link of the image in the manuscript.

Query 2:  The study states "Our study also identified two differ-360 ent SARS-CoV-2 Pangolin lineages, JN.1.1 in nasopharyngeal & saliva samples and 361 JN.1.13 in stool sample isolated on the 7th day from the onset of symptoms from the same 362 patient." and seems to imply the virus may have evolved from JN.1.1 to JN.1.13 within this patient, when it seems more likely that this was two different infections of the same patient. 

Answer 2: Thank You for pointing out the crucial finding of our study to emphasize it we have elaborated and cited more references with similar findings from line 373 to 384: SARS-CoV-2 has been shown to display location-dependent genetic variations across different anatomical regions also the genetic divergence between viral populations in the respiratory and gastrointestinal tracts. Studies attributed this to selective trans-mission events that occurred during intra-host movement, demonstrating that the genetic variability of gastrointestinal populations was higher Similarly, significant intra-host genetic variation across a variety of samples and sequencing platforms was reported, suggesting that these variations were not technical inconsistencies but rather biological differences. Furthermore, even in patients with mild symptoms, sustained viral shedding and intra-host evolution have been reported, indicating that the virus can adapt inside a single host. Together, these results provide evidence to our findings and highlight how crucial it is to take intra-host variation into account in understanding SARS-CoV-2's evolution. [48-52]

Query 3: From the methods, it is not clear to me, how stool and sewage samples are treated to potentially allow SARS-CoV-2 virus to survive to infect Vero cells while eliminating bacteria and other viruses that would contaminate the cell cultures.  The paper seems to be implying that wastewater is not a significant risk for spread of the virus, and I agree with that.  But there is possible risk of virus spread via "the fecal-oral route" not from wastewater per se but via lack of handwashing in food handlers, or similar issues.

Answer 3: Thank you for pointing out the valid concern, to eliminate any kind of contamination we have added 1% of the Penicillin-Streptomycin (10000 U/mL) solution in the minimum essential media (MEM) containing 10% Fetal Bovine serum (FBS) along with 0.25 µg/mL Amphotericin B. Revised the protocol in line 143-144.  Yes, we agree that “there is possible risk of virus spread via "the fecal-oral route" not from wastewater per se but via lack of handwashing in food handlers, or similar issues.” (Maan et al., 2022). We have detected SARS-COV 2 RNA in nasal and stool samples of COVID-19 patients and there may be possibility that the virus can spread through the stool in immunocompromised patients. Such finding highlights the importance of sanitation and disinfection to avoid possible fecal-oral spread of virus. Although in our present study no replicating virus was detected in the sewage samples, but the presence of viral RNA can be potentially used to detect the silent circulation of SAR-CoV2 in the community as degraded virus.  

Query 4: The paper discusses collecting liquid waste water, and does not mention that viral nucleic acids or perhaps whole virions can be captured from waste water on various filter material, effectively concentrating small levels of virus from much greater volumes of water. 

Answer 4: During our study, we have only used the standard protocols provided in the literature (Li T et al., 2021). Yes, we agree that “viral nucleic acids or perhaps whole virions can be captured from waste water on various filter material, effectively concentrating small levels of virus from much greater volumes of water.” There are studies suggesting the capture of Viron using microfluidic devices and nanoparticles. (Arefeh Basir et al., 2020)

Query 5: The introduction might be improved by noting that this is a positive-stranded ssRNA virus.   This is not very relevant to the pathology of the virus, but is important for understanding how to isolate and amplify the viral nucleic acid.  DNA is more durable than ssRNA, so precautions need to be used to reduce RNAse activity when isolating RNA from samples, and reverse transcription prior to PCR may be needed. 

Answer 5: Thank You for the suggestions. The changes have been made from line 36, 59-64and 141-143.
